# FORGET VECTORS AT PLAY: UNIVERSAL INPUT PERTURBATIONS DRIVING MACHINE UNLEARNING IN IMAGE CLASSIFICATION

## ABSTRACT

Machine unlearning (MU), which seeks to erase the influence of specific unwanted data from already-trained models, is becoming increasingly vital in model editing, particularly to comply with evolving data regulations like the "right to be forgotten". Conventional approaches are predominantly model-based, typically requiring re-training or fine-tuning the model's weights to meet unlearning requirements. In this work, we approach the MU problem from a novel input perturbation-based perspective, where the model weights remain intact throughout the unlearning process. We demonstrate the existence of a proactive input-based unlearning strategy, referred to *forget vector*, which can be generated as an input-agnostic data perturbation and remains as effective as model-based approximate unlearning approaches. We also show that multiple given forget vectors (*e.g.*, each targeting the unlearning of a specific data class) can be combined through simple arithmetic operations (*e.g.*, linear combinations) to generate new forget vectors for unseen unlearning tasks (*e.g.*, targeting the unlearning of an arbitrary subset across all classes). An additional advantage of our proposed forget vector approach is its parameter efficiency, as it eliminates the need for updating model weights. We conduct extensive experiments to validate the effectiveness of forget vector and its arithmetic for MU in image classification against a series of model-based unlearning baselines.

## 1 INTRODUCTION

To prevent the unauthorized use of personal or sensitive data upon completion of training and comply with legislation like "right to be forgotten" in General Data Protection Regulation (GDPR) (Hoofnagle et al., 2019), Machine Unlearning (**MU**) has gained increasing attention to tackle many *trustworthy machine learning* (ML) challenges in vision tasks, especially in image classification (Golatkar et al., 2020; Poppi et al., 2023; Warnecke et al., 2023; Fan et al., 2024). In essence, it initiates a reverse learning process to erase the impact of unwanted data (*e.g.*, specific data points, classes, or knowledge concepts) from already-trained model, while still preserving its performance and utility for information not targeted by the unlearning process. Based on the accuracy of the unlearning process and the guarantees provided regarding the removal of data from the already-trained model, existing MU methods can be roughly classified into two lines: *exact unlearning* (Dong et al., 2024; Guo et al., 2020; Thudi et al., 2022b) and *approximate unlearning* (Graves et al., 2021; Thudi et al., 2022a; Becker & Liebig, 2022; Izzo et al., 2021). Exact unlearning is the most optimal unlearning approach, ensuring the complete and verifiable removal of the targeted unwanted data. It typically involves retraining the model from scratch after excluding the data that needs to be forgotten from the original training set. However, due to its computation overhead and the lack of scalability, increasing efforts have been dedicated to the approximate unlearning manner.

Approximate unlearning offers a compromise between computational efficiency and effective data removal, making it a practical solution for many real-world scenarios. Generally, existing approximate unlearning techniques are predominantly model-based, requiring update the entire model's weights within a constrained number of training iterations to remove the influence of specific unwanted data, thereby avoiding the need for retraining the model from scratch. Among these, the representative methods include fine-tuning based approaches (Warnecke et al., 2023; Perifanis et al., 2024), gradient ascent based techniques (Thudi et al., 2022a; Chen et al., 2024) and influence function-based

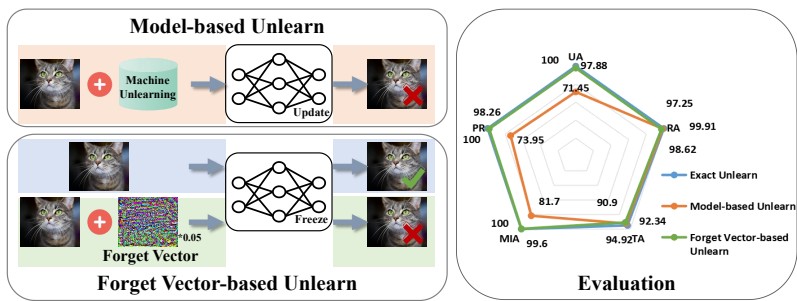

Figure 1: Left: A schematic illustration comparing our forget vector-based method, which achieves unlearning objectives without altering model parameters, to traditional model-based unlearning methods. Right: Our forget vector approach achieves the same level of unlearning performance as exact unlearning across five key metrics, while significantly outperforming model-based approximate unlearning methods over three key metrics.

methods (Golatkar et al., 2020; 2021; 2020) However, although these model-based approximate unlearning methods have achieved compelling success, they overlook the risk of degrading utility post-unlearning and typically require the involved optimization of model parameters. To alleviate the concerns on model-based approximate unlearning, we ask if it is possible append a trainable program to the input to guide the already-trained neural network for unlearning. This input-based solution is inspired by visual prompting (Bahng et al., 2022; Chen et al., 2023; Oh et al., 2023), also known as model reprogramming (Elsayed et al., 2019; Zhang et al., 2022) in transfer learning or model adaptation. For instance, the prompting method learns input perturbations to make a frozen model perform well on new tasks. These perturbations can make the model to execute tasks for which it wasn't trained. In light of this, we aim to shift the focus of MU from model-based strategies to input-based strategies, where the data is manipulated in advance with already-trained model unchanged. Above all, we raise our **key question (Q)** below:

> **(Q)** *Is there a data manipulation method that enables machine unlearning in image classification without updating model parameters? If so, how does it function, and what remarkable characteristics or properties does it possess?*

To address (Q), we advance MU through a fresh viewpoint: **forget vector**, representing a universal input perturbation designed to promote unlearning effectively and efficiently. Our key finding is that leveraging the forget vector can achieve approximate unlearning as effectively as model-based methods. Additionally, forget vector arithmetic (*e.g.*, linear combinations of multiple forget vectors) can be used to generate new forget vectors for previously unseen unlearning tasks, further enhancing its flexibility and effectiveness. See Fig. 1 for the schematic overview of our proposal and highlighted empirical performance. We summarize our contributions as follows.

• We investigate the impact of "forget data shift" (from data corruptions and adversarial perturbations) on image classifiers post-unlearning. Our findings show that unlearning is resilient to these shifts, though generalization remains vulnerable.

• Building on the resilience of machine unlearning to forget data shift, we propose a proactive, input-agnostic data perturbation strategy termed the "*forget vector*", specifically optimized to facilitate unlearning. These input-based forget vectors show comparable effectiveness to model-based unlearning methods in both class-wise and data-wise forgetting scenarios.

• We demonstrate the effectiveness of forget vector arithmetic by using class-wise forget vectors to generate new vectors that effectively remove the influence of specific data subsets in image classification models.

• We conduct extensive experiments on CIFAR-10 and ImageNet-10 to demonstrate the superiority of forget vector over various model-based unlearning baseline methods.

## 2 REVISITING MACHINE UNLEARNING AND EVALUATION

**Machine unlearning.** MU seeks to alter machine learning models and erase the impact of specific data points or classes due to privacy or copyright concerns. In terms of application areas and target

models, most MU methods currently focus on language and vision, which have garnered the greatest attention (Wang et al., 2024b). Although these two directions share the common goal of efficient data removal, techniques and challenges differ due to the distinct nature of textual and visual data.

• *MU in language models* has focused on removing the influence of specific data points, phrases, or documents from the already-trained model, adjusting the textual representations to ensure that sensitive or unwanted information is no longer retained (Shi et al., 2024; Wang et al., 2024a; 2023a; Liu et al., 2024). A novel unlearning method, SeUL, was introduced to focus on specific sequence spans rather than entire instances, which facilitates selective, fine-grained, and effective unlearning in language models (Wang et al., 2024a). Additionally, inspired by the use of weights and function-space priors to reconstruct model gradients, a recent work focuses on data removal through knowledge gap alignment and is easily generalizable to various natural language processing tasks such as classification, translation, and response generation (Wang et al., 2023a).

• *MU in vision models* has been extensively studied for both image generation task (Li et al., 2024a; Fan et al., 2024) and image classification task (Poppi et al., 2023; Liu et al., 2023). The growing use of diffusion models in generative modeling necessitates effective machine unlearning techniques to safeguard copyrights and prevent the generation of harmful content (Li et al., 2024a; Zhang et al., 2024). For example, the concept of weight saliency was introduced to guide MU, allowing models to avoid generating unwanted content while maintaining high-quality outputs for normal images (Fan et al., 2024). Besides, MU for image classification, which has significant practical applications, has also gained attention. For example, fine-tuning-based approaches incrementally update already-trained models using a modified dataset that excludes unwanted data points (Warnecke et al., 2023; Perifanis et al., 2024). Gradient ascent-based techniques reverse the impact of unwanted data by applying gradient ascent to model parameters (Thudi et al., 2022a; Chen et al., 2024). Moreover, influence unlearning methods first leverage influence functions to estimate how much a particular data point impacts the predictions and parameters of the model and then reverse those contributions (Golatkar et al., 2020; 2021; 2020). Furthermore, the connection between MU and model pruning has been explored, with findings that model sparsity helps bridge the gap between approximate and exact unlearning (Jia et al., 2023). Most existing MU methods in this domain are model-based, which can lead to utility degradation after unlearning and are often computationally expensive due to the need for model parameter updates.

• *MU in other areas* like graphs (Li et al., 2024b; Dong et al., 2024) and time-series data (Du et al., 2019) has also been explored, though the existing works are limited.

**Model adaptation Technique.** It has emerged as a promising approach to modify or repurpose already-trained models for new tasks or specific objectives without fully retraining the model. It is especially valuable for reducing computational costs and leveraging existing knowledge embedded in models. Representative branches of model adaptation technique include:

• *Visual Prompting* provides a new way to adapt already-trained models in vision by adding or modifying prompts (visual cues) in the input data to guide the model's behavior without changing its weights. For example, introducing trainable parameters in the input space while keeping the model backbone frozen can achieve comparable results with reduced computational overhead (Jia et al., 2022; Wang et al., 2023b).

• *Model Reprogramming* keeps the already-trained model fixed and modifies the input to adapt the model for different tasks. For instance, adversarial perturbations can be applied to test-time inputs to make a model perform a task chosen by the attacker (Tsai et al., 2020; Elsayed et al., 2019).

• *Feature-Based Domain Adaptation* applies transformations or mapping techniques to the input data, aligning the feature distributions between the source and target domains while keeping the model unchanged(Tahmoresnezhad & Hashemi, 2017).

To fully harness the advantages of model adaptation techniques and address the challenges in existing MU methods for image classification, we focus on machine unlearning through the design of universal input-agnostic perturbations. Our goal is to achieve comparable unlearning performance without altering the already-trained model. Notably, to the best of our knowledge, this is the first attempt to tackle machine unlearning in image classification using input-based universal perturbations.

## 3  PRELIMINARIES AND PROBLEM STATEMENT

**MU Problem Formulation.**  Let $\mathcal{D} = \{(\mathbf{x}_i, y_i)\}_{i=1}^N$ denote a training dataset consisting of $N$ instances, where $\mathbf{x}_i \in \mathbb{R}^d$ denotes the $i$-th image in $d$-dimension and $y_i \in \mathbb{R}$ refers to corresponding category label. $\mathcal{D}_f \subseteq \mathcal{D}$ stands for a subset target at erasing from already-trained model, termed as *forgetting dataset*. Accordingly, the complement of $\mathcal{D}_f$ is the *remaining dataset*, *i.e.*, $\mathcal{D}_r = \mathcal{D} \setminus \mathcal{D}_f$. Moreover, we define $\boldsymbol{\theta}$ as the model parameters, and $\boldsymbol{\theta}_o$ refer to the original model trained on the entire training set $\mathcal{D}$. Similarly, $\boldsymbol{\theta}_u$ corresponds to an unlearned model. The problem of MU lies in developing an accurate and efficient scrubbing mechanism that can effectively remove the influence of specific data points from a trained model $\boldsymbol{\theta}_o$ to $\boldsymbol{\theta}_u$. Following existing studies (Jia et al., 2023; Thudi et al., 2022a; Golatkar et al., 2020), we evaluate our proposal in the context of two classic forgetting tasks: *class-wise forgetting* where unlearning $\mathcal{D}_f$ corresponds to data points of entire class, and *random data forgetting* where unlearning $\mathcal{D}_f$ consists of a subset randomly selected from all classes (In our experiments, we randomly select a certain proportion of data from the dataset across all classes, such as "%10"). In the inference phase, we have an original *testing dataset*. Notably, in the scenario of *class-wise forgetting*, we split original testing dataset into two groups: $\mathcal{D}_t$ and $\mathcal{D}_{ft}$, which means the remaining dataset and forgetting set in the original testing set, respectively. As for the random data forgetting case, the whole testing set is considered as $\mathcal{D}_t$.

**Representative MU Methods.**

• Retrain: This is the most optimal (exact) MU method, where the model is retrained from scratch using the remaining dataset $\mathcal{D}_r$. However, this approach imposes a significant computational cost, especially when training deep neural networks (DNNs) on large-scale datasets. Despite its inefficiency, Retrain serves as the benchmark for MU, representing the ideal result that other MU methods aim to achieve.

• Fine-tuning (**FT**) (Warnecke et al., 2023; Golatkar et al., 2020): Instead of retraining the model from scratch, FT fine-tunes the already-trained model $\boldsymbol{\theta}_o$ on $\mathcal{D}_f$ for a few iterations to obtain $\boldsymbol{\theta}_u$. This approach balances computational efficiency and effective data removal, making it a practical alternative to exact unlearning, especially for large models and datasets.

• Random Label (**RL**) (Golatkar et al., 2020): To reduce the influence of specific data points or classes, RL intentionally corrupts the labels of the data in $\mathcal{D}_f$ by randomly assigning new labels, thereby reducing the impact of $\mathcal{D}_f$ on the model's learned representations.

• Gradient Ascent (**GA**) (Graves et al., 2021): GA adjusts the parameters of the already-trained model $\boldsymbol{\theta}_o$ in a specific direction to reverse the learning associated with the data in $\mathcal{D}_f$.

**Evaluation Metric.** To comprehensively characterize the proposed scheme in MU, we employed several commonly used evaluation metrics following prior approaches (Jia et al., 2023; Thudi et al., 2022b) to comprehensively assess the effectiveness of data removal from different aspects. Here we chose five metrics as follows.

• Unlearning Accuracy (**UA**): To assess the efficacy of MU in terms of accuracy, we define $\mathrm{UA}(\boldsymbol{\theta}_u) = 1 - \mathrm{Acc}_{\mathcal{D}_f}(\boldsymbol{\theta}_u)$, where $\mathrm{Acc}_{\mathcal{D}_f}(\boldsymbol{\theta}_u)$ denotes the classification accuracy of $\boldsymbol{\theta}_u$ on the forgetting dataset $\mathcal{D}_f$. Essentially, the smaller the gap between the approximate unlearning method and exact unlearning method (**Retrain**), the better the performance of the machine unlearning (MU) method.

• Remaining Accuracy (**RA**): To reflect the fidelity of MU, we test the accuracy of $\boldsymbol{\theta}_u$ on $\mathcal{D}_r$.

• Testing Accuracy (**TA**): To assess how well $\boldsymbol{\theta}_u$ retains its generalization capabilities on the testing data $\mathcal{D}_t$ after the unlearning procedure, we also report the accuracy on $\mathcal{D}_t$.

• Membership inference attack (**MIA-Efficacy**): We use MIA to evaluate the performance of MU from an alternative perspective, where a confidence-based MIA predictor (Song et al., 2019) is applied to $\boldsymbol{\theta}_u$ on $\mathcal{D}_f$. Numerically, MIA indicates the success rate that the data points in $\mathcal{D}_f$ can be successively identified as the forgetting samples of $\boldsymbol{\theta}_u$. Details about MIA can be found in this work Jia et al. (2023).

• Predictive robustness (**PR**): To evaluate the UA performance of MU regarding new unseen forgetting data $\mathcal{D}_{ft}$, we also introduce the concept of predictive robustness. In practice, in the context of class-wise forgetting, $\mathcal{D}_{ft}$ can be directly obtained from the original testing set. In terms of the random-data

Table 1: The performance comparison of already-trained model and one exact unlearning method (Retrain) and three approximate unlearning methods (FT, RL, and GA) on CIFAR-10 and ImageNet-10 datasets with respect to four unlearning evaluation metrics on benign images and images with different data shift scenarios. w/Corruption 1 and w/Corruption 2 refers to adding two different level of Gaussian noise to the benign images, while w/Adv1 and w/Adv2 denotes two setting of PGD attack. It is worth noting that we only report the result of one trail where the random seed is set as 1.

| Dataset | Model | Method | Benign | | | | w/ Corruption 1 | | | | w/ Corruption 2 | | | | w/ Adv 1 | | | | w/ Adv 2 | | | |
|---|---|---|---|---|---|---|---|---|---|---|---|---|---|---|---|---|---|---|---|---|---|---|
| | | | UA↑ | RA↑ | TA↑ | MIA↑ | UA↑ | RA↑ | TA↑ | MIA↑ | UA↑ | RA↑ | TA↑ | MIA↑ | UA↑ | RA↑ | TA↑ | MIA↑ | UA↑ | RA↑ | TA↑ | MIA↑ |
| | | | | | | | | | | | **Class-wise Forgetting** | | | | | | | | | | | |
| CIFAR-10 | ResNet-18 | Origin | 0.00 | 100 | 94.64 | 0.20 | 59.20 | 45.30 | 43.40 | 48.90 | 83.76 | 25.08 | 24.43 | 84.90 | 89.22 | 12.99 | 11.21 | 83.80 | 98.64 | 1.12 | 1.07 | 98.20 |
| | | Retrain | 100 | 100 | 95.12 | 100 | 100 | 39.62 | 38.36 | 100 | 100 | 23.78 | 0.00 | 100 | 100 | 26.56 | 24.19 | 100 | 100 | 4.60 | 4.50 | 0.00 |
| | | FT | 100 | 92.55 | 88.99 | 100 | 100 | 48.39 | 47.76 | 100 | 100 | 30.75 | 30.26 | 100 | 100 | 30.60 | 28.91 | 100 | 100 | 8.11 | 8.09 | 0 |
| | | RL | 15.48 | 99.98 | 94.67 | 99.99 | 79.84 | 46.18 | 44.36 | 47.50 | 97.40 | 26.02 | 26.14 | 71.70 | 95.28 | 14.06 | 12.17 | 86.30 | 98.74 | 1.08 | 1.13 | 1.80 |
| | | GA | 71.86 | 98.69 | 92.34 | 81.80 | 99.96 | 35.48 | 34.29 | 100 | 100 | 31.60 | 21.43 | 0.00 | 100 | 18.30 | 16.44 | 100 | 96.80 | 0.00 | 3.34 | 0.00 |
| ImageNet-10 | VGG-16 | Origin | 0.15 | 99.94 | 97.11 | 3.80 | 16.77 | 92.71 | 90.67 | 22.60 | 40.62 | 83.37 | 81.78 | 72.80 | 35.69 | 82.86 | 78.67 | 32.4 | 98.15 | 11.53 | 8.67 | 99.20 |
| | | Retrain | 100 | 99.92 | 97.56 | 100 | 100 | 92.06 | 91.33 | 100 | 100 | 84.50 | 83.11 | 100 | 100 | 84.65 | 81.33 | 100 | 100 | 16.62 | 14.44 | 0.00 |
| | | FT | 40.53 | 99.74 | 97.33 | 55.1 | 81.77 | 88.18 | 85.33 | 85.20 | 94.15 | 73.52 | 71.11 | 93.20 | 86.77 | 83.87 | 80.22 | 89.50 | 100 | 16.22 | 12.44 | 100 |
| | | RL | 86.77 | 98.77 | 96.22 | 93.50 | 96.85 | 88.68 | 87.11 | 0.40 | 98.77 | 77.33 | 75.78 | 0.20 | 97.93 | 82.75 | 77.78 | 98.90 | 100 | 18.95 | 19.33 | 0.00 |
| | | GA | 97.77 | 94.54 | 90.22 | 98.50 | 99.62 | 82.22 | 79.56 | 99.80 | 100 | 58.20 | 57.56 | 100 | 100 | 76.14 | 75.56 | 100 | 100 | 22.98 | 23.11 | 100 |
| | | | | | | | | | | | **Random Data Forgetting (10%)** | | | | | | | | | | | |
| CIFAR-10 | ResNet-18 | Origin | 0.00 | 100 | 94.77 | 0.30 | 57.26 | 42.79 | 41.05 | 51.10 | 77.76 | 22.84 | 22.32 | 87.00 | 86.42 | 12.68 | 10.92 | 79.60 | 98.94 | 1.15 | 1.06 | 98.40 |
| | | Retrain | 5.28 | 100 | 94.58 | 12.30 | 68.34 | 33.63 | 32.50 | 62.00 | 83.48 | 17.25 | 17.24 | 18.40 | 87.44 | 13.56 | 11.65 | 81.10 | 98.66 | 1.35 | 1.19 | 97.70 |
| | | FT | 0.40 | 99.79 | 94.08 | 2.40 | 58.06 | 42.80 | 41.43 | 60.90 | 76.22 | 24.94 | 24.42 | 15.70 | 85.50 | 14.57 | 12.71 | 78.80 | 97.96 | 1.93 | 1.94 | 1.20 |
| | | RL | 5.26 | 98.96 | 91.91 | 12.20 | 60.90 | 41.13 | 39.10 | 62.00 | 77.54 | 23.42 | 22.94 | 69.60 | 86.08 | 15.81 | 13.50 | 77.00 | 97.12 | 3.16 | 3.03 | 94.80 |
| | | GA | 2.40 | 97.93 | 93.43 | 4.50 | 57.60 | 42.59 | 41.04 | 52.20 | 76.54 | 24.58 | 24.50 | 85.50 | 83.52 | 15.89 | 14.27 | 77.00 | 98.42 | 1.55 | 1.53 | 1.40 |
| ImageNet-10 | VGG-16 | Origin | 0.15 | 99.94 | 97.00 | 2.80 | 8.92 | 91.84 | 89.60 | 12.30 | 20.23 | 81.10 | 80.40 | 21.90 | 19.92 | 81.11 | 77.20 | 14.80 | 89.15 | 10.53 | 8.40 | 92.60 |
| | | Retrain | 2.69 | 99.32 | 98.00 | 90.80 | 40.77 | 60.40 | 58.20 | 43.60 | 52.00 | 49.16 | 47.20 | 25.70 | 43.92 | 58.93 | 54.80 | 39.10 | 87.04 | 13.59 | 14.00 | 89.50 |
| | | FT | 0.92 | 99.59 | 97.20 | 5.20 | 12.69 | 88.26 | 87.00 | 16.10 | 24.14 | 75.39 | 75.80 | 82.20 | 20.54 | 79.65 | 76.40 | 25.80 | 88.54 | 12.80 | 11.20 | 89.80 |
| | | RL | 1.46 | 99.57 | 97.40 | 3.40 | 13.77 | 88.39 | 85.60 | 41.50 | 25.69 | 76.22 | 75.40 | 59.70 | 25.85 | 79.91 | 76.80 | 47.60 | 85.39 | 15.54 | 14.60 | 85.50 |
| | | GA | 0.15 | 99.92 | 97.20 | 2.80 | 8.62 | 91.91 | 89.00 | 12.50 | 20.23 | 81.06 | 80.20 | 18.20 | 19.69 | 81.05 | 77.40 | 15.70 | 89.00 | 11.00 | 8.60 | 91.70 |

forgetting scenario, we customize a new $\mathcal{D}_{ft}$ through introducing a certain degree of corruption to the data in $\mathcal{D}_f$.

# 4 FORGET DATA SHIFT ON IMAGE CLASSIFIERS POST-UNLEARNING

To adapt already-trained model for specific tasks, methods like fine-tuning and linear probing have gained significant attention (Seo et al., 2024; Huang et al., 2024), though both require access to the model and have drawbacks such as computational overhead and overfitting risks. Reently, input transformation techniques, including image-to-image translation (Murez et al., 2018), visual prompting (Oh et al., 2024), and adversarial reprogramming (Elsayed et al., 2019), have emerged as alternatives, and achieved similar levels of performance. Inspired by their success and the impressive results obtained, we aim to explore a proactive input-based unlearning strategy. Before diving into this topic, we first explore the impact of forget data shift on image classifiers post-unlearning and observe how MU handle such introduced shift.

**Forget Data Shift.** In our work, we define "data shift" as explicitly altering the distribution of the data through data corruptions and adversarial perturbations. The details of these two kinds of data shift manners are as follows.

• **Data Corruptions**. To benchmark the robustness of classifiers to common perturbations, 15 diverse corruption types applied to validation images of ImageNet Deng et al. (2009) are designed in (Hendrycks & Dietterich, 2019), where corruptions are drawn from four main categories, *i.e.*, noise, blur, weather, and digital. Each type of corruption has five levels of severity, and the higher the severity, the larger the noise factor. In our work, we apply Gaussian noise to image with severity level $h = 1$ or 2, namely, "w/Corruption 1" and "w/Corruption2". Formally, Gaussian noise (GN) from symmetric Gaussian distribution $\mathcal{N}(0, c^2\boldsymbol{I})$ with 0 mean vector and $d$ by $d$ covariance matrix $c^2\boldsymbol{I}$. $c = 0.12 \times h - 0.04$ in our settings, and perturbed data $\mathbf{x}_i^{'}$ are generated by summing up $\mathbf{x}_i$ with GN.

• **Adversarial Perturbations**. An adversarial image is a benign image that has been modified with a precisely crafted small distortion aimed at misleading a classifier. These subtle perturbations can occasionally deceive black-box classifiers (Dabkowski & Gal, 2017). In our work, we adopt the Projected Gradient Descent (PGD) attack (Deng & Karam, 2020) to generate adversarial examples through iterative gradient updates, rendering the attacked images incapable of being correctly classified by the respective models. Specifically, for each iteration $t$, we have, $\mathbf{x}_{t+1}^{'} = \Pi_{B(\mathbf{x}, \epsilon)} \left( \mathbf{x}_t^{'} + \alpha \cdot \text{sign} \left( \nabla_{\mathbf{x}} \mathcal{L}(\boldsymbol{\theta}, \mathbf{x}_t^{'}, y) \right) \right)$, where $\mathcal{L}$ is the loss function of the model with parameters $\boldsymbol{\theta}$ and true label $y$, $\nabla_{\mathbf{x}} \mathcal{L}(\boldsymbol{\theta}, \mathbf{x}_t^{'}, y)$ is the gradient of the loss with respect to the input $\mathbf{x}$. Meanwhile, $\alpha$ is the step size, and $B(\mathbf{x}, \epsilon)$ is the $\epsilon$-ball around $\mathbf{x}$. In our work, we employ two

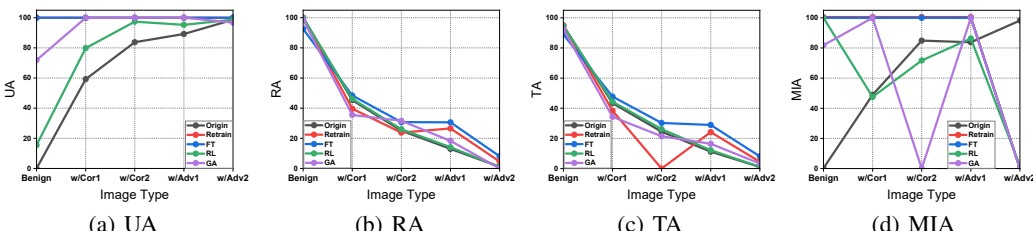

| (a) UA | (b) RA | (c) TA | (d) MIA |

Figure 2: Influence of different data shift strategies on four evaluation metrics for the CIFAR-10 dataset using the ResNet-18 network in the class-wise forgetting scenario.

parameter settings, namely, "w/Adv1" ($\alpha$ equals to $2/255$, $\epsilon$ is $8/255$ and iteration is 7) and "w/Adv2" ($\alpha$ equals to $0.01$, $\epsilon$ is $0.3$ and iteration is $40$).

**Finding of Forget Data Shift**. To comprehensively observe the impact of data shift to the Original model $\theta_o$ and existing model-based MU methods, we reported the results of four different metrics in Table 1. As can be seen, we present the performance of the original model, the exact UM (Retrain), and three approximate UM methods (FT, RL, GA) on benign images, images with two types of data corruptions, and adversarial perturbations applied, across different dataset splits. To better analyze the performance trends of different methods across various metrics, we also provide corresponding curve plots regarding the class-wise forgetting setting on CIFAR-10 using ResNet18 network architecture in Figure 2. For both Table 1 and Figure 2, we can draw the following observations:

❶ *Better Unlearning* on the forgetting dataset $\mathcal{D}_f$. When data corruption and adversarial perturbation are added to the benign images, whether for the original model $\theta_o$, exact unlearning retrain $\theta_r$, or other UM methods $\theta_u$, the unlearning capability of the model tends to improve. One possible explanation is that the added corruption or perturbation can distort or obscure the critical features that the model has learned to recognize(Original) or disregard (Retrain and approximate Unlearn), making it harder for the model to retain its original knowledge. Meanwhile, as the level of corruption or the strength of the attack increases, the improvement in the model's unlearning performance also tends to grow. This likely happens because more severe corruptions or stronger adversarial attacks distort the input data more significantly, making it harder for the model to retain previously learned information, thereby enhancing its ability to "forget." In other words, the more aggressive the perturbation, the greater the disruption, leading to more effective unlearning. In fact, this phenomenon is totally opposite to the motivation and expectations of MU, whose goal is to maintain high TA and RA, demonstrating the model's ability to correctly classify data other than ones needed to be forgotten.

❷ *Worse Post-unlearning* on the remaining dataset($\mathcal{D}_r$) and testing dataset ($\mathcal{D}_t$). Due to the introduction of data corruption or adversarial perturbation, the classification accuracy on the remaining dataset and the testing dataset significantly decreases for all the methods. Whether it is data corruption or adversarial perturbation, as the degree of data shift increases, the model's post-unlearning performance deteriorates. This is clearly demonstrated by the fact that the introduction of data corruption and adversarial perturbation significantly impairs the classification capabilities retained by both the original model and the UM models on the remaining dataset as well as the testing dataset.

❸ *Different Sensitivity to Corruptions*. Notable, regrading the MIA value, different methods behave differently. For example, as shown in Fig 2(d), as data corruption and adversarial perturbation are introduced, the original model becomes unable to accurately distinguish whether the forget set was seen during training process, and original model struggle to maintain such distinction, causing MIA values to rise. However, for the MU methods, there seems to be no uniform trend. One potential reason is that different unlearning methods have varying levels of robustness against data corruption. Methods that are more sensitive to corruptions might lose their ability to effectively distinguish between training and forget sets, while more robust methods may maintain such distinction.

Above all, we found how these data shifts affect the model's ability to forget designated data while maintaining classification accuracy on the remaining dataset. Our findings reveal that, although the unlearning mechanisms generally demonstrate resilience to these forget data shifts, successfully forgetting the specified data, the model's ability to generalize remains vulnerable. Such vulnerability may present serious challenges to directly introducing a type of corruption or perturbation to the dataset in the context of MU. To this end, our work aims to explore a data manipulation method and design an input-based unlearning scheme with already-trained model intact, where an input-agnostic

data perturbation can be generated to meet the unlearning requirement and preserve the image classifiers post-unlearning ability as existing model-based approximate MU methods, simultaneously.

# 5    FORGET VECTORS: UNIVERSAL INPUT PERTURBATIONS FOR MU

In this section, we present the proposed input-based unlearning strategy, referred to *forget vector*, as the major novelty. This method addresses the MU problem by generating an input-agnostic data perturbation that is as effective as model-based unlearning techniques. We first establish the input-agnostic perturbation strategy, then explore the potential to create new forget vectors for unseen data subsets by combining forget vectors designed for specific classes. Concurrently, we conduct an analysis and comparison to determine what kind of loss function would be most advantageous for our proposed approach.

**MU via Forget Vector.** Suppose that there is an input-agnostic pixel-wise perturbation $\delta$ having the same dimensions as the input data. This perturbation could be updated iteratively via gradient-based optimization, allowing to learn and refine the perturbation toward the direction that maximally enhances both unlearning and post unlearning performance. Since $\delta$ is input-agnostic, it is applied uniformly across all data, affecting each data point in the same way. Formally, for each instance in forgetting dataset $\mathcal{D}_f = \{(\mathbf{x}_i, y_i)\}_{i=1}^{|D_f|}$, we have the perturbed one $\mathcal{D}'_f = \{(\mathbf{x}_i + \delta, y_i)\}_{i=1}^{|D_f|}$, where $y_i$ is corresponding category label of image $\mathbf{x}_i$. As for the remaining dataset and testing dataset, $\mathcal{D}_r$ and $\mathcal{D}_t$, in a similar manner, we can derive $\mathcal{D}'_r = \{(\mathbf{x}_j + \delta, y_j)\}_{j=1}^{|D_r|}$ and $\mathcal{D}'_t = \{(\mathbf{x}_k + \delta, y_k)\}_{k=1}^{|D_t|}$. Furthermore, it is essential to highlight that our input-based $\delta$ is optimized using the already-trained model, without making any changes to the model's parameters. All optimization steps are carried out on the fixed model, focusing exclusively on refining the input perturbation.

In a sense, as to better preserve the unlearning capability of already-trained model on forgetting dataset while maximizing its retained performance on the training dataset, we design distinct objective functions for them, each tailored to their respective optimization goals. Specifically, during the optimization process of the target $\delta$, we adopt the C&W untargeted attack loss (Carlini & Wagner, 2017) for the forgetting set after comparing with other loss function like Random Label-based Cross-Entropy Loss, aiming to classify the "perturbed data" into any label other than the ground truth one. Formally, we have loss term as follows,

$$\mathcal{L}_{atk}(\mathbf{x} + \delta; D_f) = \sum_{\mathbf{x} \in D_f} \max\{f_{y_i}(\mathbf{x} + \delta) - \max_{t \neq y_i} f_t(\mathbf{x} + \delta), -\tau\}, \tag{1}$$

where $y_i$ denotes the ground-truth label of data $\mathbf{x}$, and $t$ represents the logit value of class $t$ with respect to the perturbed data $\mathbf{x}'$. Here, $\tau \geq 0$ is a given constant used for characterize the attack confidence. Meanwhile, as for the remaining dataset, we utilize the Cross-Entropy Loss (Mao et al., 2023) as follows,

$$\mathcal{L}_{CE}(\mathbf{x} + \delta; D_r) = - \sum_{\mathbf{x} \in D_r} \sum_{i=1}^{C} y_i \log(\hat{y}_i(\mathbf{x} + \delta)), \tag{2}$$

where $C$ is the number of category labels, and $\hat{y}$ is the predicted probability distribution of perturbed data $\mathbf{x}'$.

Consequently, we have the following objective function towards our designed forget vector,

$$\mathcal{L} = \alpha \mathcal{L}_{atk}(\mathbf{x} + \delta; D_f) + \beta \mathcal{L}_{CE}(\mathbf{x} + \delta; D_r) + \lambda ||\delta||_2^2, \tag{3}$$

where $\alpha$, $\beta$, and $\lambda$ are the nonnegative tradeoff parameters, and $|| \cdot ||_2^2$ denotes the Euclidean norm.

Above all, we generate the optimization problem to find a universal perturbation that meets our desired requirements as follows,

$$\min_{\delta} \alpha \mathcal{L}_{atk}(\mathbf{x} + \delta; D_f) + \beta \mathcal{L}_{CE}(\mathbf{x} + \delta; D_r) + \lambda ||\delta||_2^2. \tag{4}$$

**Compositional Unlearning**. In our work, we address two types of unlearning modes: class-wise forgetting and random data forgetting. To gain deeper insights, we further explore a new task scenario,

termed *compositional unlearning* to generate universal input perturbations from a new perspective, where class-specific forget vectors are modified and combined through arithmetic operations (*e.g.*, linear combinations) to generate a new forget vector for an unseen unlearning task involving an arbitrary subset across all classes. Intuitively, addressing the random data forgetting problem through compositional unlearning offers several key benefits. On the one hand, a complete re-optimization process of input perturbation is no longer needed and the optimization process only involves a limited number of weights, proving a quick solution when encountering new random data forgetting task. In a sense, computational cost is significantly reduced, making it suitable for large-scale data and models. On the other hand, the compositional approach provides robustness to shifts in data distribution, as it can dynamically adjust forget vectors based on the specific requirements of new unlearning tasks. This adaptability ensures that the model remains resilient even when data characteristics change over time. Suppose that we have obtained $K$ class-specific perturbation set $\boldsymbol{\delta} = \{\boldsymbol{\delta}_1, \boldsymbol{\delta}_2, \ldots, \boldsymbol{\delta}_K\}$, where $\boldsymbol{\delta}_i$ corresponds to $i$-th class of dataset. Adopting the compositional unlearning to acquire the target forget vector $\boldsymbol{\delta}_c$ for arbitrary random data forgetting, we have $\boldsymbol{\delta}_c = \sum_{i=1}^{K} w_i \boldsymbol{\delta}_i$, where $W = \{w_1, w_2, \ldots, w_K\}$ is the parameters that we aim to learn during the compositional unlearning optimization process. Accordingly, the optimization in Eqn. 4 can be derived into the form as follows,

$$\min_W \alpha \mathcal{L}_{atk}(\mathbf{x} + \boldsymbol{\delta}_c; D_f) + \beta \mathcal{L}_{CE}(\mathbf{x} + \boldsymbol{\delta}_c; D_r) + \lambda ||W||_2^2. \tag{5}$$

## 6 EXPERIMENTS

### 6.1 EXPERIMENT SETUPS.

**Datasets and classifiers** To evaluate the performance of our designed input-based perturbation, we focus on the image classification under CIRAR-10 (Krizhevsky et al., 2009) using ResNet-18 (He et al., 2016) and ImageNet-10 (Deng et al., 2009) adopting VGG-16 (Simonyan & Zisserman, 2015) model architecture. Notably, ImageNet-10 is carefully selected from specific coarse-grained classes of the ImageNet-1K (Deng et al., 2009) dataset, taking into account the diversity and breadth of the dataset. More details regarding these two datasets can be found in the Appendix A1.

**Evaluation metrics.** As discussed in Section 3, we adopt five metrics, where UA and MIA reflect the efficacy of MU, RA depicts the fidelity of MU, and TA characterize the generalization ability of unlearning method. Different from existing works, we also introduce a new testing setting PR to evaluate the predictive robustness of the unlearning methods facing the new unseen forgotten data. We follow the (Jia et al., 2023) regarding the implementation of the MIA, and details in terms of MIA can be found in this work (Jia et al., 2023). Note that a smaller performance gap with Retrain indicates better performance of an MU method. To provide an overall assessment, the metric *Averaging (Avg.) Gap* is also introduced and calculated as the average of the performance gaps measured in accuracy-related metrics, including UA, RA, TA and MIA.

**Parameter setting.** In our work, we focus on two unlearning scenarios: *class-wise forgetting* and *random data forgetting*. For simplicity, we randomly select one specific class from the CIFAR-10 and ImageNet-10 datasets to verify the effectiveness of the designed forget vector, respectively. Simultaneously, for the random data forgetting scenario, we set the forget ratio at $10\%$. To avoid the randomness of results, both our method and baseline methods were tested 10 times with different random seeds. In the real implementation, we fixed the parameters of original already-trained model and the targeted perturbation $\delta$ was initialized to zero. In general, large extensive experiments demonstrate that the influence of $\tau$ is minor. Hence, in our work, we set it as $1$. It indicates that we start with no modification to the input data, and the optimization process will then gradually adjust the perturbation from this neutral starting point to achieve the desired effect. Pertaining to the optimization, we utilized the stochastic gradient descent (SGD)Amari (1993) with the momentum factor as $0.9$. The grid search strategy was adopted to determine the optimal values for parameters (*i.e.*, $\alpha$, $\beta$, $\lambda$ and $\tau$). If not otherwise specified, we reported the best performance with the parameters optimized to their best values. Furthermore, we set the batch size to be $256$ for both two datasets using two model networks.

Table 2: Performance overview of various Machine Unlearning (MU) methods for image classification under 10% random data forgetting, on CIFAR-10 and ImageNet-10 using both ResNet-18 and VGG-16. Results are reported in the format $a_{\pm b}$, where $a$ is the mean and $b$ denotes standard deviation $b$ over 10 independent trials. The performance gap against Retrain is indicated in (•). Meanwhile, *PR* indicates the predictive robustness of forgetting ability facing new unseen data that is similar to the data intended to be forgotten. And the performance of our proposal is shown in boldface.

| Dataset | Model | Method | Random Data Forgetting(10%) | | | | | |
| --- | --- | --- | --- | --- | --- | --- | --- | --- |
| | | | UA↑ | RA↑ | TA↑ | MIA↑ | *Avg.Gap*↓ | *PR*↑ |
| CIFAR-10 | ResNet-18 | Retrain | $5.50_{\pm 0.16}(0.00)$ | $99.88_{\pm 0.05}(0.00)$ | $94.24_{\pm 0.19}(0.00)$ | $11.57_{\pm 0.47}(0.00)$ | 0.00 | 63.97 |
| | | FT | $0.03_{\pm 0.03}(5.47)$ | $99.98_{\pm 0.02}(0.10)$ | $94.45_{\pm 0.14}(0.21)$ | $0.75_{\pm 0.09}(10.82)$ | 4.15 | 64.56 |
| | | RL | $0.52_{\pm 0.24}(4.98)$ | $99.85_{\pm 0.07}(0.03)$ | $93.88_{\pm 0.20}(0.36)$ | $3.13_{\pm 0.55}(8.44)$ | 3.45 | 60.00 |
| | | GA | $1.56_{\pm 3.08}(3.94)$ | $98.67_{\pm 2.74}(1.21)$ | $92.84_{\pm 2.59}(1.40)$ | $2.88_{\pm 3.44}(8.69)$ | 3.81 | 57.6 |
| | | **Ours** | $\mathbf{2.61_{\pm 0.49}}(2.89)$ | $\mathbf{97.33_{\pm 0.47}}(2.55)$ | $\mathbf{90.97_{\pm 0.38}}(3.27)$ | $\mathbf{8.26_{\pm 1.17}}(3.00)$ | 2.92 | 66.89 |
| ImageNet-10 | VGG-16 | Retrain | $4.05_{\pm 0.45}(0.00)$ | $99.48_{\pm 0.07}(0.00)$ | $96.33_{\pm 0.38}(0.00)$ | $6.60_{\pm 1.07}(0.00)$ | 0.00 | 40.77 |
| | | FT | $1.35_{\pm 0.32}(2.70)$ | $99.36_{\pm 0.28}(0.12)$ | $96.54_{\pm 0.59}(0.21)$ | $4.67_{\pm 1.61}(1.93)$ | 1.24 | 12.85 |
| | | RL | $2.96_{\pm 0.42}(1.09)$ | $99.19_{\pm 0.16}(0.29)$ | $95.50_{\pm 0.9}(0.83)$ | $12.85_{\pm 4.25}(6.25)$ | 2.12 | 16.15 |
| | | GA | $0.18_{\pm 0.04}(3.87)$ | $99.86_{\pm 0.01}(0.38)$ | $97.47_{\pm 0.09}(1.14)$ | $2.97_{\pm 1.51}(3.63)$ | 2.23 | 8.92 |
| | | **Ours** | $\mathbf{2.27_{\pm 0.50}}(1.78)$ | $\mathbf{98.29_{\pm 0.32}}(1.19)$ | $\mathbf{95.82_{\pm 0.48}}(0.51)$ | $\mathbf{6.13_{\pm 1.40}}(0.47)$ | 0.99 | 17.83 |

Table 3: Performance overview of various Machine Unlearning (MU) methods for image classification under the scenario of class-wise forgetting on CIFAR-10 and ImageNet-10 using ResNet-18 and VGG-16, respectively. The reporting format is the same as Table 2, and the performance of our proposal is shown in boldface.

| Dataset | Model | Method | Class-wise Forgetting(class 1) | | | | | |
| --- | --- | --- | --- | --- | --- | --- | --- | --- |
| | | | UA↑ | RA↑ | TA↑ | MIA↑ | *Avg.Gap*↓ | *PR*↑ |
| CIFAR-10 | ResNet-18 | Retrain | $100.00_{\pm 0.00}(0.00)$ | $99.91_{\pm 0.03}(0.00)$ | $94.92_{\pm 0.15}(0.00)$ | $100.00_{\pm 0.00}(0.00)$ | 0.00 | 100 |
| | | FT | $5.27_{\pm 0.73}(94.73)$ | $100.0_{\pm 0.0}(0.09)$ | $95.03_{\pm 0.07}(0.11)$ | $51.49_{\pm 5.07}(48.51)$ | 35.86 | 21.44 |
| | | RL | $18.87_{\pm 7.34}(81.13)$ | $99.98_{\pm 0.0}(0.07)$ | $94.51_{\pm 0.12}(0.41)$ | $98.94_{\pm 0.79}(1.06)$ | 20.67 | 27.99 |
| | | GA | $71.45_{\pm 0.35}(28.55)$ | $98.62_{\pm 0.04}(1.29)$ | $92.34_{\pm 0.02}(2.58)$ | $81.7_{\pm 0.22}(18.3)$ | 12.68 | 73.95 |
| | | **Ours** | $\mathbf{97.88_{\pm 0.27}}(2.12)$ | $\mathbf{97.25_{\pm 0.24}}(2.66)$ | $\mathbf{90.90_{\pm 0.32}}(4.02)$ | $\mathbf{99.60_{\pm 0.15}}(0.40)$ | 9.20 | **98.26** |
| ImageNet-10 | VGG-16 | Retrain | $100.00_{\pm 0.00}(0.00)$ | $99.66_{\pm 0.16}(0.00)$ | $97.11_{\pm 0.82}(0.00)$ | $100.00_{\pm 0.00}(0.00)$ | 0.00 | 100 |
| | | FT | $39.66_{\pm 4.73}(60.34)$ | $99.78_{\pm 0.03}(0.13)$ | $97.27_{\pm 0.35}(2.35)$ | $55.76_{\pm 7.26}(44.24)$ | 26.77 | 33.00 |
| | | RL | $76.58_{\pm 11.64}(23.42)$ | $99.28_{\pm 0.2}(0.63)$ | $96.91_{\pm 0.55}(1.99)$ | $46.04_{\pm 33.71}(53.96)$ | 20.00 | 76.60 |
| | | GA | $46.61_{\pm 6.11}(53.39)$ | $99.35_{\pm 0.11}(0.56)$ | $95.6_{\pm 0.22}(0.68)$ | $49.15_{\pm 9.36}(50.85)$ | 26.37 | 40.40 |
| | | **Ours** | $\mathbf{87.23_{\pm 6.55}}(12.77)$ | $\mathbf{94.77_{\pm 1.16}}(5.14)$ | $\mathbf{94.04_{\pm 1.29}}(0.88)$ | $\mathbf{91.41_{\pm 5.9}}(8.59)$ | 6.85 | **87.60** |

## 6.2 EXPERIMENT RESULTS

**Forget vector improved approximate unlearning**. To comprehensively evaluate the introduced *forget vector*, we first reported the various MU performance in image classification of the most optimal (exact) MU method (Retrain), three representative approximate MU approaches (FT, RL, and GA) , and our proposed one in Tables 2 and 3. For these two tables, we can draw the following observations. ❶From the perspective of performance gap against Retrain, our proposed forget vector based MU method consistently outperforms all the other approximate MU approaches under both the class-wise forgetting and random data forgetting scenarios. In particular, with the best baseline, our forget vector based method achieves the improvement of 0.53, 0.25, 3.48, and 13.15 in both tasks on two datasets and network architectures, respectively. Notably, such improvement is directly benefit from the introduction of the forget vector when feeding inputs into the model, with the already-trained model intact. The possible explanation is that our designed forget vector alter how the data is represented in the model and modify the feature space representation. Meanwhile, the already-trained model is sensitive to such input change and its dependency on specific data is weaken, where the output behavior of the model is no longer relied on the certain data and unlearning can be achieved without model parameter update. ❷Meanwhile, for both two forgetting scenarios, we also evaluate the predictive robustness (PR) of forgetting ability when facing the new unseen data that is similar to initial forgotten data. As can be seen from the last column of Tables 2 and 3, overall, the forgetting ability of our proposal is significantly better than all the approximate MU methods. Taking the class-wise forgetting on CIFAR-10 for example, our proposal achieves the unlearning accuracy 98.36, largely higher than the best baseline GA (under the PR metric) whose forgetting accuracy is 73.95. In this way, we can state that our studied forget vector is resilient and robust to the targeted forgetting elements. ❸ Interestingly, from a broader perspective, we can find that our proposal demonstrates superior forgetting capabilities, although the retaining accuracy and testing

Table 4: Performance overview of the compositional unlearning on different data forgetting ratios, where RD denotes the initial random-data forgetting case (*e.g.*, learn the forget vector for a specific subset from scratch) and LC refers to the compositional unlearning from pre-learned forget vectors for each data class.

| Dataset | Model | Forgetting Ratio | Method | UA↑ | RA↑ | TA↑ | MIA↑ | *Avg.Gap* |
|---|---|---|---|---|---|---|---|---|
| CIFAR-10 | ResNet-18 | 10% | Retrain | $5.50_{\pm0.16}(0.00)$ | $99.88_{\pm0.05}(0.00)$ | $94.24_{\pm0.19}(0.00)$ | $11.57_{\pm0.47}(0.00)$ | 0.00 |
| | | | RD | $2.61_{\pm0.49}(2.89)$ | $97.33_{\pm0.47}(2.55)$ | $90.97_{\pm0.38}(3.27)$ | $8.26_{\pm1.17}(3.00)$ | 2.92 |
| | | | LN | $5.36_{\pm0.60}(0.14)$ | $94.93_{\pm0.64}(4.95)$ | $88.60_{\pm0.59}(5.64)$ | $9.76_{\pm0.91}(1.81)$ | 3.16 |
| | | 50% | Retrain | $7.95_{\pm0.17}(0.00)$ | $99.59_{\pm0.82}(0.00)$ | $91.78_{\pm0.22}(0.00)$ | $16.68_{\pm0.68}(0.00)$ | 0.00 |
| | | | RD | $70.54_{\pm0.81}(62.59)$ | $29.88_{\pm0.97}(69.71)$ | $29.72_{\pm0.79}(62.06)$ | $41.78_{\pm24.71}(25.1)$ | 54.87 |
| | | | LN | $60.92_{\pm2.59}(52.97)$ | $38.17_{\pm2.32}(61.42)$ | $38.05_{\pm2.28}(53.73)$ | $24.5_{\pm1.4}(7.82)$ | 43.99 |
| ImageNet-10 | VGG-16 | 10% | Retrain | $4.05_{\pm0.45}(0.00)$ | $99.48_{\pm0.07}(0.00)$ | $96.33_{\pm0.38}(0.00)$ | $6.60_{\pm1.07}(0.00)$ | 0.00 |
| | | | RD | $2.27_{\pm0.50}(1.78)$ | $98.29_{\pm0.32}(0.51)$ | $95.82_{\pm0.48}(0.51)$ | $6.13_{\pm1.40}(0.47)$ | 0.99 |
| | | | LN | $2.27_{\pm1.18}(0.00)$ | $97.93_{\pm1.09}(0.36)$ | $91.41_{\pm1.25}(4.41)$ | $4.95_{\pm1.86}(1.18)$ | 1.49 |
| | | 50% | Retrain | $5.65_{\pm0.30}(0.00)$ | $99.23_{\pm0.14}(0.00)$ | $94.56_{\pm0.73}(0.00)$ | $16.72_{\pm24.45}(0.00)$ | 0.00 |
| | | | RD | $5.57_{\pm0.49}(3.30)$ | $95.36_{\pm0.64}(2.93)$ | $93.38_{\pm0.65}(2.44)$ | $12.65_{\pm1.71}(6.52)$ | 3.29 |
| | | | LN | $1.87_{\pm1.05}(3.78)$ | $98.28_{\pm0.86}(0.95)$ | $96.33_{\pm0.77}(1.77)$ | $4.70_{\pm1.87}(12.02)$ | 4.63 |

accuracy, compared to the baselines, is relatively acceptable. In a sense, this phenomenon has great application potential in scenarios where the priority for forgetting requirements is higher like the protection of user information privacy from being disclosed is crucial. ❹As a key takeaway, we highlight that training a forget vector with the same dimensions as the image (*e.g.*, $224 \times 224 \times 3$ for ImageNet-10 dataset) can significantly enhance the computational efficiency compared to fine-tuning the entire weights of deep neural network (*e.g.*, 138 million parameters of VGG-16 network).

**Compositional unlearning: an efficient approach**. To gain more deep insight, we further investigate the performance of compositional unlearning in the context of our "forget vector". Here, we chose the initial random-data forgetting approach. As can be seen from Table 4, the introduced compositional unlearning through the simple linear combination of the pre-learned class-specific forget vectors consistently comparable to the initial random-data forgetting case, except for the 50% random-data forgetting in CIFAR-10 using ResNet-18, where both two solutions do not work. In a sense, one possible reason is thatthe CIFAR-10 dataset has relatively low-resolution images, making it more sensitive to input-based perturbations. When the amount of data to forget increases, for instance, to 50%, the model may increasingly struggle to differentiate between perturbed and benign data as training progresses, leading to a decline in performance. In contrast, such situation does not exist in the ImageNet-10 dataset.

**Component Analysis**. To verify the effectiveness of each key component in Eqn.3, we investigated the sensitivity of our proposed forget vector to them. Through extensive experiments, we discovered that when both $\alpha$ and $\lambda$ are set to 1, adjusting the value of $\beta$ can balance the performance of **RA** and **UA**, where the parameter can be selected according to specific forgetting priority requirements. Specifically, we take class-wise forgetting on CIFAR-10 with ResNet-18 as an example. In this case, we first fix the values of $\alpha$ and $\lambda$ at 1, then modify the $\beta$ value to demonstrate the sensitivity analysis of $\beta$. The results are displayed in Figure 3.

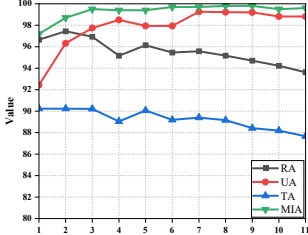

Figure 3: Sensitivity analysis on four evaluation metrics in terms of the hyper-parameter $\beta$.

## 7 CONCLUSION

In this paper, we focus on studying the problem of machine unlearning in image classification from a new perspective, referred to forget vector. Unlike existing model-based machine unlearning methods where the retraining or fine-tuning of the model's weights are required, our proposal demonstrates the existence of input-agnostic data perturbation. Notably, our designed strategy remains as effective as model-based approximate machine unlearning approaches. Interestingly, we also verify that new vectors for unseen learning tasks such as the unlearning of an arbitrary subset across all classes can be generated through the simple arithmetic operations like linear combination of pre-obtained forget vectors of specific class. In a sense, benefit from the parameter efficiency of such compositional unlearning, new unlearning tasks can be solved in a more efficient manner. Extensive experiments have been conducted on two datasets using two different network architectures and the results demonstrate the effectiveness of the proposed scheme and validate the benefits of our optimized forget vector.

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

APPENDIX

# A  ADDITIONAL EXPERIMENTAL DETAILS AND RESULTS

## A.1  DATASETS AND MODELS

Statistics regarding our datasets are listed in Table A1. And the carefully selected 10 classes of ImageNet dataset can be found in Table A2.

Table A1: Summary of the CIFAR-10 and ImageNet-10.

|              | CIFAR-10 | ImageNet-10 |
|--------------|----------|-------------|
| Training Set | $50,000$ | $13,000$    |
| Testing Set  | $10,000$ | $500$       |
| Labels       | $10$     | $10$        |

Table A2: 10 category name of ImageNet-10.

| Dataset      | Class name     |                 |                |             |             |
|--------------|----------------|-----------------|----------------|-------------|-------------|
| ImageNet-10  | tabby cat      | Siberian husky  | American robin | convertible | airliner    |
|              | mountain bike  | schooner        | daisy          | strawberry  | grand piano |

