# OpenReview forum: "Forget Vectors at Play: Universal Input Perturbations Driving Machine Unlearning in Image Classification"
_ICLR.cc/2025/Conference — ICLR 2025 Conference Withdrawn Submission_

### Official Review · Reviewer_bvme · 2024-10-31

**Soundness:** 3
**Presentation:** 3
**Contribution:** 3
**Rating:** 5
**Confidence:** 4

**Summary:**

This paper proposes a novel input perturbation-based method for machine unlearning. It generate an input-agnostic data perturbation named forget vector using an optimization problem, requiring no model parameter tuning. The experiment results demonstrate its effectiveness compared to model-based MU methods.

**Strengths:**

The method is the first attempt to use input-based MU method, offering new insight.
It is easy to understand.
Computational overhead of input-based MU is much less than model-based MU.

**Weaknesses:**

For method: Although the input-based method is efficient compared to model-based methods, I think the core of MU is to let the model forget   specific information instead of deceiving the model to predict wrong information on forget data.  Therefore, somehow I think the paper do not achieve real forgetting.

For experiment: The results in Table 2 and Table 3 show that the test accuracy of input-based MU is much lower than other methods, which is a serious drawback of input noise based method. Moreover, the prominent advantage of input-based MU is reflected in MIA metric while other metrics are comparable or even worse than model-based methods. Maybe the reason is that MIA is sensitive to input noise. This is easy to understand because a sample with noise is more likely to be identified as out-of-distribution data when comparing to a clean sample. Therefore, the performance of input-based MU is open to discussion.

**Questions:**

see weakness

---

### Official Review · Reviewer_HvW2 · 2024-11-03

**Soundness:** 3
**Presentation:** 3
**Contribution:** 3
**Rating:** 5
**Confidence:** 4

**Summary:**

Instead of model-based MU approach, this paper proposes a perturbation-based method called forget vectors. Forget vectors are shown to be as effective as previous model unlearning, with better parameter efficiency, and more flexibilities by leveraging vector properties (for random data unlearning). The motivation seems to be preventing the performance degradation of the model that is been unlearned.

**Strengths:**

1. Sec 4 in insightful, establishing preliminary experiment and evaluation settings, as well as gaining fundamental observations.

2. The idea is novel to me, which I interpret as applying learnable mask vectors to the data. The intuition also aligns with recent studies on low-rank learning.

3. The paper writing is good, easy to follow.

**Weaknesses:**

1. The CIFAR-10+ResNet18 and IN-10+VGG16 have roughly the same parameters/data pixels ratio (e.g., for the first setting: 11M/(32x32x3x50k)$\approx$0.07). But they can have different behaviors since Table 1, why? Is it more data-dependent? It also makes me curious about more over-parameterized settings as well as larger scale experiments, as I am not sure if a generalizable conclusion across or related to scales (in terms of model and data size) can be drawn.

2. I find compositional unlearning insightful which may benefit future work, but the experiments are lacking. It is really crucial to demonstrate the compositional unlearning's ability under distribution shift, e.g., applying learned vectors from one data domain to another, so that the authors can prove the claims. From the given equations, I am unsure about whether combining $K$ trained class-specific perturbation vectors in one domain can apply to another data domain effectively. Are the dimension of $w_i$ 1? If so, that is even worse because we only tune $K$ parameters for shifting domain, which lacks theoretical guarantees on its learnability. Maybe comparing three settings: 1) fine-tuning to another domain, 2) training forget vectors in another domain from scratch, 3) zero-shot, can help us analyze these aspects, and it might also depends on how far two domains are.

**Questions:**

Please refer to the weakness section, where I expect two sets of experiments, and weakness 2) is more important than 1) to me. I am concerned about whether 1) this approach scales up, 2) this approach is not domain-specific. The idea is good, I find it interesting, thanks for the hard work.

---

### Official Review · Reviewer_TiLm · 2024-11-03

**Soundness:** 3
**Presentation:** 3
**Contribution:** 2
**Rating:** 5
**Confidence:** 5

**Summary:**

The proposed testing approach for Machine Unlearning (MU), referred to as "Forget Vectors," introduces an input-based perturbation strategy to achieve effective data removal without altering the model's parameters. Instead of conventional model-based retraining, Forget Vectors provide a parameter-efficient, scalable approach that applies universal perturbations to inputs, achieving data unlearning while maintaining model performance on non-forgotten data.

**Strengths:**

•	Parameter Efficiency: Forget Vectors circumvent the need for parameter updates, significantly reducing computational requirements compared to model retraining or fine-tuning.
•	Novel Unlearning Objective: The modified objective function introduces a perspective on adversarial examples, extending their utility to enable unlearning.
•	Preserves Model Integrity: By focusing on input perturbations, the approach retains the model’s original weights, thus ensuring model integrity and reducing risks associated with full retraining.

**Weaknesses:**

•	GDPR Compliance Concerns: The paper’s reliance on approximate unlearning without theoretical guarantees presents a significant shortfall. While approximate unlearning may be practical, it falls short in scenarios where data privacy and regulatory compliance are non-negotiable. Without provable guarantees, it is questionable whether this method can satisfy GDPR requirements for data erasure. This gap undermines the core purpose of Model Unlearning in privacy-centered contexts, where the "right to be forgotten" demands more than a probabilistic assurance.
•	Scalability to Other Domains: The Forget Vector approach is developed and validated primarily for image classification tasks, potentially limiting its application in NLP or other non-visual domains where input perturbations may be less effective.
•	Dependence on MIA (Membership Inference Attack) Testing via Ulira: While the paper uses MIA testing as a metric for unlearning effectiveness, the effectiveness of MIA testing itself is not sufficiently robust for privacy guarantees. Additionally the use of U-LiRA [1] is recommended.
•	Sensitivity to Data Shifts: From the paper the effectiveness of unlearning decreases under certain data shifts, which may hinder the reliability of Forget Vectors in dynamic data environments or adversarial settings.

**Questions:**

1.	How might the Forget Vector approach be adapted or expanded to suit domains beyond image classification, such as NLP or time-series data?
2.	Are there any potential risks of unintended consequences (e.g., degradation of model utility) when applying compositional unlearning for arbitrary data subsets? Can an adversary compromise the utility by choosing the "right" subset?

[1] Kurmanji, Meghdad, Peter Triantafillou, Jamie Hayes, and Eleni Triantafillou. "Towards unbounded machine unlearning." Advances in neural information processing systems 36 (2024).

---

### Official Review · Reviewer_iqGW · 2024-11-03

**Soundness:** 2
**Presentation:** 2
**Contribution:** 2
**Rating:** 3
**Confidence:** 4

**Summary:**

In this paper, the authors introduce an input perturbation technique called the "forget vector," which enables machine unlearning (MU) without altering model weights. The authors claim that the Forget vectors are versatile and can be combined through arithmetic operations to unlearn new, unseen tasks, such as removing specific subsets of data across classes. With experiments in image classification, they have shown that the proposed approach is parameter-efficient, significantly reducing the need for model reconfiguration while maintaining MU effectiveness.

**Strengths:**

Strengths:
* The idea of forgetting vectors for machine unlearning is new and interesting.
* The main idea of the paper is clearly presented in most parts.

**Weaknesses:**

Weakness:
* The paper claims to address class forgetting and random sample forgetting in machine unlearning. It is not clear if it can address the 'sample unlearning' presented in [1]. Experiments and comparisons with the baseline are recommended to show such unlearning.
* The paper did not cite and compare with many established methods in MU area. For example, Deep Unlearning [2] performs MU without iterative fine-tuning or retraining. This and similar methods are not cited as related works and are not compared in the paper.
* The scale of the experiments is very limited. In this paper authors only showed unlearning when the model is trained on up to 10 classes. However, [2] already showed that class unlearning is possible with 1k classes from the Imagenet dataset. It is not clear if the proposed method is scalable to such a large number of classes or not.
* No calculation, discussion, and comparison of computational cost are presented in the paper.


[1]Meghdad Kurmanji, Peter Triantafillou, and Eleni Triantafillou. Towards unbounded machine unlearning.
arXiv preprint arXiv:2302.09880, 2023.
[2] Kodge, S., Saha, G., Roy, K.: Deep unlearning: Fast and efficient training-free
approach to controlled forgetting. arXiv preprint arXiv:2312.00761 (2023) . https://openreview.net/pdf?id=BmI5p6wBi0

**Questions:**

* It is not clear how Compositional Unlearning works. Please revise this part to improve the clarity of the paper.
* Can the method unlearn some 'specific samples' from the pre-trained model?
* Provide a comparison with SoTA works (see above).  I suggest experiments and comparisons when random samples, specific samples, and a specific class are unlearned from the 1k ImageNet dataset.

---

### Official Review · Reviewer_u2rZ · 2024-11-04

**Soundness:** 3
**Presentation:** 2
**Contribution:** 3
**Rating:** 6
**Confidence:** 2

**Summary:**

The paper introduces an innovative approach to machine unlearning (MU) by focusing on input perturbations, using "forget vectors". They can facilitate unlearning without modifying the model’s parameters, which contrasts with traditional model-based unlearning methods that often require retraining or fine-tuning. The forget vectors are designed as input-agnostic perturbations, applicable across different data instances.

**Strengths:**

- The paper presents a unique shift from model-based MU techniques to an input-perturbations methodology. This approach does not involve updating model weights, making the approach computationally efficient.
- The ability to create new forget vectors through arithmetic combinations of pre-learned vectors showcases the method’s flexibility and adaptability.

**Weaknesses:**

- While unlearning is highlighted as important, the paper would benefit from demonstrating the method in a real-world application, such as mitigating bias or removing harmful data to better showcase its utility.
- The experiments are limited to small datasets and simpler models. It would be insightful to understand how this method performs on more complex datasets and larger-scale models.
- Although the focus is on image classification, the paper could discuss potential extensions or insights into how this approach might generalize to other tasks.
- There are some minor typo issues in lines 243, 405, and 310.

**Questions:**

See weaknesses above.

---

### Note · Authors · 2024-11-15

I have read and agree with the venue's withdrawal policy on behalf of myself and my co-authors.